# Effect of Animal Stocking Density and Habitat Enrichment on Survival and Vitality of Wild Green Shore Crabs, *Carcinus maenas,* Maintained in the Laboratory

**DOI:** 10.3390/ani12212970

**Published:** 2022-10-28

**Authors:** Charlotte H. Wilson, Russell C. Wyeth, John I. Spicer, Iain J. McGaw

**Affiliations:** 1Department of Ocean Sciences, Memorial University, 0 Marine Lab Road, St. John’s, NL A1C 5S7, Canada; 2School of Biological and Marine Sciences, University of Plymouth, Drake Circus, Plymouth PL4 8AA, Devon, UK; 3Department of Biology, St. Francis Xavier University, 4130 University Avenue, Antigonish, NS B2G 2W5, Canada

**Keywords:** animal welfare, crab, decapod, health, protocol, time

## Abstract

**Simple Summary:**

Decapod crustaceans are used extensively in laboratory experiments. Recently in the United Kingdom decapods have been included in animal care protocols. However, little is known about how captive conditions affect the survival and general condition of wild crustaceans. We used the green shore crab, *Carcinus maenas,* to investigate the effects of stocking density and shelter on survival and vitality indices during a 6 month period in the laboratory. Both stocking density and the presence of shelter did not affect survival, nor have a clear effect on vitality indices (limb loss, claw strength, ‘blood’ protein concentration, righting time, leg flare and retraction). However, vitality indices showed that crab condition was declining over time: this became most apparent after 8 to 11 weeks storage in the laboratory. This decline in condition was likely due to repeated handling of the crabs, rather than the stocking conditions, which apparently led to a cumulative stress and a deterioration in animal health. Bringing wild crustaceans into the laboratory and holding them, even with modest experimental manipulation, may result in high mortality rates. Researchers and animal care committees need to be aware that wild captive invertebrates will respond very differently to laboratory-bred vertebrates, and plan experiments accordingly.

**Abstract:**

The wide geographic distribution, large size and ease of capture has led to decapod crustaceans being used extensively in laboratory experiments. Recently in the United Kingdom decapod crustaceans were listed as sentient beings, resulting in their inclusion in animal care protocols. Ironically, little is known about how captive conditions affect the survival and general condition of wild decapod crustaceans. We used the green shore crab, *Carcinus maenas,* to investigate the effects of stocking density and shelter on survival and vitality indices during a 6 month period in the laboratory. Neither stocking density nor the presence of shelter affected survival. Stocking density also had no effect on the vitality indices (limb loss, claw strength, BRIX, righting time, leg flare and retraction). The presence of shelter did affect the number of limbs lost and the leg retraction response, but had no effect on the other vitality indices. All vitality indices changed, and mortality increased over time, independent of treatment: this became most apparent after 8 to 11 weeks storage in the laboratory. This decline in condition may have been due to repeated handling of the crabs, rather than the stocking conditions. In support of this, untracked, non-handled (control) individuals sustained a 4% mortality rate compared with 67% mortality in experimental crabs during the 6 month period. Although simple experimental monitoring of crabs with biweekly vitality tests only produced transient short-term stress events, the repeated handling over time apparently led to a cumulative stress and a deterioration in animal health. Bringing wild crustaceans into the laboratory and holding them, even with modest experimental manipulation, may result in high mortality rates. Researchers and animal care committees need to be aware that wild captive invertebrates will respond very differently to laboratory-bred vertebrates, and plan experiments accordingly.

## 1. Introduction

Animals have been used in scientific research for centuries, both in the field and in captive conditions. It is commonly believed that captivity increases life expectancy and reduces stress for some species compared to their wild counterparts, but this has rarely been tested [1]. For many species, captivity is far from ideal and can sometimes cause stress and suffering to the animal (reviewed in [1,2]). Animals may need to be held captive in the laboratory environment for a prolonged time period due to seasonal field sampling constraints or long-term experimental designs. Given the sheer volume of laboratory work using marine animals, notably in the context of environmental change [3,4], the welfare implications of keeping marine animals in laboratory conditions for long-term periods has increasing pertinence [5,6].

The global ubiquity, vast taxonomic diversity [7] and relative ease of captive husbandry of decapod crustaceans make them desirable for use in both in situ and ex situ experiments, and therefore they have been employed extensively in research related to environmental change [8,9]. Decapod crustaceans also have substantial economic importance in fisheries and the aquaculture industry, prompting further research on selected species [10,11,12].

Despite arguably being better able to withstand capture, transport and storage than other aquatic animals due to their durable exoskeleton, and in some cases the ability to breathe in air, decapods still experience a substantial amount of stress during the above practices [13,14]. Researchers have investigated optimal transport and storage conditions, specific to commercial practices, in an attempt to maximise product quality and animal welfare [13,15,16]. However, work documenting the influence of laboratory and research settings on crustacean welfare is still sparse.

The seemingly unnatural conditions associated with the laboratory environment may present a plethora of stressors if not regulated appropriately. It is accepted that mimicking a natural environment is the most appropriate way to mitigate stress in crustaceans, but only limited quantification or evidence for the success of this exists [15,17]. Whilst efforts are made to ensure environmental stability in a laboratory environment, many stressors are unavoidable. They may include, but are not limited to, bright lighting, noise, excessive handling and inappropriate stocking densities [15,18,19,20,21,22]. Recent work highlights concern with keeping Atlantic rock crab (*Cancer irroratus*) in the laboratory for prolonged holding times (3 months), as indicated by a change in metabolism and haemolymph protein levels [23]. However, experiments recording changes over even longer time periods and assessing the impact of methods used to mitigate the laboratory stressors listed above are lacking.

The measurement of acute stress responses is relatively well documented for decapod crustaceans: oxygen consumption rate [24,25,26] crustacean hyperglycaemic hormone and haemolymph glucose concentrations [27,28] and haemolymph L-lactate concentrations [26,29,30] are widely used to quantify crustacean responses to many environmental and physical (e.g., handling) stressors. Other haemolymph parameters, e.g., BRIX and haemolymph protein density are an increasingly credible, cost and time effective way to indicate the physiological condition of large crustaceans [31,32,33]. More subjective measures, e.g., behavioural and reflex impairment (RAMP) indicators [34] are also established as a relatively reliable way to predict mortality in many taxa, including crustaceans [35,36,37]. 

The majority of decapod species used in laboratory experiments are still harvested from the wild and maintained in the laboratory prior to use in tests and experiments. Apart from a small number of commercially cultured species, little is known about their general condition during laboratory maintenance or even their survival. These data, however, have become increasingly important as awareness of decapod welfare has vastly increased, with multiple review articles emerging as a result [5,14,38,39]. Many of these reviews, the primary literature therein, and subsequent commentaries), e.g., [40] suggest the need for increased animal welfare protection for large crustaceans, e.g., [41,42,43,44]. Until recently, only the welfare of vertebrate species and cephalopods required animal care protocols when being used in scientific research in most western countries (e.g., *Animals (Scientific Procedures) Act 1986, UK).* Guided by a recent independent review [45], the UK government were first to legally recognise decapod crustaceans as sentient beings (Animal Welfare (Sentience) Bill, 2021), setting a benchmark for animal welfare protocols. Such recognition is now spreading across Europe and is likely to be implemented by other nations. This will likely have far-reaching impacts on the use of decapods in scientific research, and further highlights the need to understand best practices for maintaining healthy animals in captive conditions [42,43,46].

The green shore crab (*Carcinus maenas*, Linnaeus, 1758) is described as a ‘hardy’ decapod species, resilient to many environmental stressors (reviewed in [47,48]). Its relatively large size, ease of capture and the fact that it has been listed in the top 100 worst invasive species have made it one of the most studied crabs in the world [47]. Indeed, it has been referred to colloquially as the “marine biologist’s rat” because it has been used in numerous ecological, physiological and toxicological studies. Therefore, data for *C. maenas* can provide a baseline for responses to long-term laboratory captivity, from which optimal storage conditions for other decapod species could be refined. In particular, their study will help determine possible ways to reduce, replace and refine (3R’s) numbers of decapods used in experiments; practices that are commonly applied to other research animals [49].

The primary aim of this study was to use behavioural and physiological measures to observe changes in vitality, physical condition and survival of *C. maenas* when maintained in the laboratory for a prolonged period (6 months). Given that captive crustaceans typically exhibit much greater mortality rates than lab-bred vertebrates [14,50] this study will provide important baseline information on the condition and longevity of wild crustaceans held in laboratory settings. In turn, this will also improve the quality and real-life accuracy of research using these animals.

## 2. Material and Methods

### 2.1. Animal Collection

Intermoult male green crabs, *C. maenas*, (50–75 mm carapace width), were harvested by trapping during September 2018 at Fox Harbour, Newfoundland, Canada. Female crabs were not retained due to permitting issues associated with holding this invasive species in captivity. Crabs were transferred to the Department of Ocean Sciences, Memorial University (St. Johns, NL, Canada), and initially kept in a holding tank (Volume = 3000 L) continuously supplied with flow through sea water (Temperature = 10–12 °C, Salinity = 31–32) on a natural day-night light regime. The water was aerated using air stones (>85% oxygen saturation) and crabs were fed a mixed diet of fish, mussels and kelp twice weekly. PVC tubes were added to provide shelter and reduce aggressive interactions. Crabs were maintained in these conditions for at least two weeks prior to use in the experiment. The tank was drained at twice weekly intervals, and any lost appendages and uneaten food remains removed. This feeding and cleaning regime for the holding tank containing spare crabs was maintained throughout the experiment.

### 2.2. Experimental Design

This experiment was designed to determine if and how stocking density and environmental enrichment (provision of shelter), influenced the condition and survival of laboratory-kept crabs. Before the experiment began, crabs were removed from the holding tank, their carapace width measured and any appendage loss was noted. A coloured foam tag was secured using cyano-acrylate adhesive to the dorsal carapace surface and numbered for identification and tracking. Only crabs with a carapace width of 50–75 mm, possessing both claws, and with no more than two missing walking legs were used in experiments. Crabs were not separated by the colour of their underside (which can influence some physiological responses), but rather a continuum of colours from green through yellow-orange to red was used across all treatments [51,52].

Tagged crabs were placed into separate, plastic-coated, wire mesh cages (30 cm × 15 cm × 30 cm deep; 1 cm^2^ mesh), stocked at one of three different densities in each cage: 1 tagged crab only (total = 1), 1 tagged crab with 3 additional unmarked crabs (total = 4) and 1 tagged crab with 7 additional unmarked crabs (total = 8). Furthermore, to determine if environmental enrichment affected the crab’s condition, cylindrical PVC shelters (10 cm diameter × 20 cm length) were added to half of the cages. In all, 144 tagged crabs were followed over a period of 27 weeks (24 tagged crabs in each of the six treatments: 1, 4, or 8 crabs with shelter and 1, 4, or 8 crabs without shelter). The additional unmarked crabs were not handled, but their survivorship was recorded and therefore acted as a control group for assessing mortality in response to handling. Unmarked crabs played no role in the assessment of vitality. Cages from all the 6 treatment groups were evenly distributed across three separate flow-through aerated seawater tanks (same water source, Volume = 3000 L, Temperature = 10–12 °C, Salinity = 31–32, natural day-night light cycle). Crabs were fed a mixed diet of fish, mussels and kelp to excess twice weekly. The tanks were drained and rinsed weekly to remove any uneaten food and debris. The experiment began in late September 2018 and was terminated in late March 2019: this ensured that the crabs would not moult during this time [53]. The crabs were checked twice weekly and any mortalities were recorded and removed. If any limbs were lost by the tagged crabs, the number and position of the lost limbs were recorded.

### 2.3. Vitality Indices

In addition to recording mortalities, several vitality indices (listed below), were measured at regular intervals for tagged crabs only. If a tagged crab died during the experiment, they were not replaced. If an unmarked crab died, it was replaced with another of a similar size from the original holding tank, to maintain original density. Before measurements were taken, a tagged crab was removed from its cage and placed in a bucket containing sea water from their original tank. By the nature of these tests, most had to be performed in air, but no individual crab was emersed for greater than 30 min. The following indices were measured every two weeks.

Chelal compression strength was used as an indicator of muscle strength and competitive fitness [54,55,56]. This measure was both a function of the physical ability of the crab, but also its willingness to engage the muscles, i.e., it is not a measure of strength alone but a composite of strength and aggressive propensity [22]. Crabs were removed from their container and encouraged to exert a force on a digital hanging scale (Brecknell Electrosampson) using their dominant (crusher) chelae. The scale was mounted onto a clamp stand and secured using cable ties. A secured, stationary latch was present at the base of the clamp, approximately 0.5 cm distally from the hook of the digital scale [57]. The bottom (fixed finger) of the crab’s claw (propodus) was placed into the secured latch, whilst the dactyl was placed into the hook attached to the scale. Crabs were agitated by tapping their carapace until they pinched the claw together, subsequently pulling the upper hook of the scale. Crabs were given three opportunities to provide a reading. If no response was observed, their ‘strength’ was recorded as absent data. Any crabs with broken or missing claws were excluded from further analyses. Because claw crushing force positively correlates with size (maximum height of propodus) [58], a strength index was calculated by dividing compression strength (kg) by claw propodus height (mm). 

Righting behaviour is described by Shirley and Stickle [59] as a “complex reflex requiring muscle coordination and neurological control that can be a sensitive measure of well-being.” Hence, a reduction in righting ability can indicate a decline in health and subsequent mortality. For measurement of righting behaviour, individuals were placed into a small (30 cm diameter) bucket, containing sea water from their original tank. They were left to settle in the bucket for two minutes and then manually inverted and placed back on the base of the bucket, ventrum up. Righting time was recorded as time taken to return to a dorsum up position. This process was repeated three times in succession and the fastest time to right was recorded and used for statistical analysis. Crabs that did not right within 120 s were recorded as unable to right and returned to their cage. Righting time was affected by both the number of limbs lost and their specific location (pers. obs.). Therefore, if a crab had lost more than three limbs on one side of the body, or both of the last (fourth pair) walking legs, it was not used in subsequent analyses to ensure that righting time remained an indicator of vitality rather than a function of limb loss.

Several reflex impairment (RAMP) [34] indicators are used as a relatively reliable way to predict mortality in many taxa, including crustaceans [35,36,37]. We identified two reflex responses as being consistently useful indicators of health in *C. maenas*: leg flare and leg retraction. For leg flare, a crab was gently picked up by gripping either side of the carapace, which would often cause it to flare all limbs out horizontally. This response was classified as strong, weak or absent (see Table 1).

If the crab lost its fourth pair (back legs) of walking legs, or more than three walking legs on one side the response could not be observed properly, and this crab was no longer used. For leg retraction, the crab was held in a similar manner, the first pair of walking legs were drawn backward with the handler’s forefinger and the degree of opposing force recorded (Table 1). If a crab lost both of the first walking legs, or more than three walking legs on one side, it was not used in subsequent leg retraction analyses.

In addition to these behavioural indicators, the BRIX (Bx) levels (amount of soluble solids in a liquid) of the haemolymph were measured once every four weeks. Haemolymph BRIX levels are directly correlated with haemolymph protein levels and have been used as an indicator of nutritional or moult status [32,33,60]. It also correlates with the mass of the heart, hepatopancreas and muscles, and therefore can be used as an indicator of physiological condition or vitality [31]. For measurement, approximately 500 µL of haemolymph was extracted from the infrabranchial sinus via the arthrodial membrane at the base of a walking leg (legs changed every four weeks) using an 18-gauge needle and syringe. The haemolymph was immediately transferred onto the sample well of a pre-calibrated Brix/RI Chek Digital Pocket Refractometer (Reichert Analytical Instruments, Depew, NY, USA) and a reading was obtained) [31]. Time between extracting the haemolymph and processing the sample did not exceed 30 s.

### 2.4. Statistical Analysis

The mortality rates of crabs in the six different treatments, as well as the overall mortality rates for the tagged and untracked crabs, were analysed using a Kaplan–Meier Log rank survival test. The effects of stocking density and shelter on limb loss and vitality indices over time were tested using a non-parametric analysis for longitudinal data (nparLD), [61]. Analysis was carried out in R version 2022.7.1.554 (R Core Team, 2022) [62], using RStudio [63] and the nparLD version 2.2 package. In all cases, the nparLD ANOVA-type test statistics were used from a full factorial model with main effects for time, shelter and density and the various interactions. As the nparLD procedure can also be used for ordinal data, the leg flare and leg retraction responses were assigned numerical scores for analysis (3 = strong, 2 = weak, 1 = none). A consequence of mortality during the experiment was an increase in missing data values over time, precluding post hoc comparisons between time points following the nparLD procedure. Furthermore, because the main effects of shelter and density were rarely significant, while the main effect of time was always significant, we chose to pool data from all treatments at each time point. Thus, to determine where differences occurred over time (and regardless of treatment) in each vitality index, we used these pooled data to plot bootstrapped 95% confidence intervals (using the ggplot2 package version 3.3.6) [64] and when these intervals did not overlap this was considered a significant difference between time points. 

In addition to testing for effects over time, analyses were carried out to compare animals that died during the experiment with those that remained alive at the end of the 27week period. The final number of limbs lost for each crab was compared between crabs that died during the experiment with those that survived the 27 week period (combining all six treatments) using a Mann–Whitney U test. As there was no significant difference in limb loss when comparing the crabs that remained alive with those that died, final limb loss for all of the crabs in each of the six treatments was analysed using a one-way or rank-transformed ANOVA (SigmaStat 4.0). 

The magnitude change in claw strength and haemolymph BRIX levels was calculated as the difference between the initial and the final measurement taken from each crab. If a crab died during the experiment the last reading prior to death was used. The data for crabs that died were initially compared with those that survived until the end of the experiment (all treatments combined), using a Student’s *t*-test. Because there was a significant difference in magnitude change (for both claw strength and BRIX) between crabs that died during the experiment and those that survived, these two groups (dead and alive) were further analysed separately for the six treatments using one-way or rank-transformed ANOVAs (SigmaStat 4.0).

## 3. Results

### 3.1. Survival

There was a steady mortality rate beginning after approximately four weeks of storage in the lab tanks (Figure 1A). Mean survival times ranged between 19.4 and 23.6 weeks for each treatment (Table 2). 

The survival times of the crabs were not affected by the density or the presence of a shelter (Kaplan–Meier Log Rank Survival test, F_5_ = 4.459, *p* = 0.485). Overall, 67% of the tagged crabs died during the 27 week experimental period. In contrast the mortality of the untracked crabs was significantly lower (Figure 1B); less than 4% of these crabs died during the experiment (Kaplan–Meier Log Rank Survival test, F_1_ = 316.36, *p* < 0.001). 

### 3.2. Vitality Indices

There was no significant effect of stocking density on any of the vitality indices (Table 3). The presence of a shelter did affect individual limb loss and the leg retraction response, but there were no significant effects of shelter on the other vitality indices (Table 3). There was, however, a significant change in all indices with time, with an overall deterioration in crab condition the longer they were maintained in the laboratory. 

### 3.3. Limb Loss

The presence of a shelter affected the number of limbs lost by crabs during the course of the experiment (Figure 2A). In particular, at the two lower densities (one and four), crabs with a shelter had lower individual limb loss than those at corresponding densities without shelter. There was no effect of shelter when comparing crabs held at the highest density. These patterns were supported by a statistically significant main effect of shelter with an interaction effect between shelter and density (Table 3). Irrespective of shelter and density treatments, there was a significant increase in limb loss during the first two weeks of the experiment. Subsequent limb loss was comparatively low between 2 and 10 weeks, after which there was a steady and continual increase in the cumulative number of limbs that individual crabs lost (treatments combined) (Figure 8A). 

The total number of limbs lost was also compared between crabs that died during the experiment and those that remained alive (all six treatments combined) to determine if limb loss was an indicator of impending mortality: there was no significant difference between the two groups (Mann–Whitney U test, T = 4014, *p* = 0.426). Indeed, several crabs that lost no limbs died during the experiment, while one crab that lost all limbs (10) was still alive at week 27. When comparing the total number of limbs lost across all six treatments (Figure 2B) (combining crabs that died during the experiment and those that remained alive within each treatment), there were some significant differences (One-way ANOVA on ranks, H_5_ = 17.75, *p* = 0.003). Single crabs without a shelter lost more limbs than single crabs with a shelter (Tukey *p* < 0.05) and eight crabs with a shelter (Tukey *p* < 0.05; there were no significant differences among the other treatments (Tukey *p* > 0.05). 

### 3.4. Claw Strength Index

Apart from a transient decrease at two weeks (which was likely due to an equipment error; Figure 3A), the overall claw strength index remained unchanged with mean values of 1.7 ± 0.1 and 2.3 ± 0.1 units (Figure 8B) across all treatments during the first eight weeks in captivity. Thereafter, a significant decline occurred at 11 weeks, with a further decline at 27 weeks. 

The change in claw strength index was also examined (for all treatments combined) for the crabs that died during the experiment and those that remained alive at the end of the experimental period (Figure 3B). Comparison of the two groups detected an overall decrease in claw strength index, decreasing by −0.27 ± 17 units prior to death, and −0.88 ± 18 units in crabs that remained alive at the end of the experiment; this difference was significant (T-test, T_132_ = −2.39, *p* = 0.018. Therefore, the six treatment groups were analysed separately for crabs that died and those that lived. In both cases there was no significant effect of treatment type on change in claw strength index (one way ANOVA, dead F_5_ = 2.29, *p* = 0.056; alive F = 2.35, *p* = 0.052). 

### 3.5. BRIX

The overall initial mean BRIX values were 8.4 ± 1.4 °Bx; these increased during the next three monthly sampling periods (weeks 4–12). Thereafter, they declined to levels that were similar to initial measurements (Figure 4A and Figure 8C). 

The change in BRIX (between initial level and final reading) was also analysed for crabs that survived the entire period and separately for those that died (all six treatments combined) (Figure 4B). The crabs that died exhibited a significant decrease in BRIX by 0.62 ± 0.23 °Bx, whereas those crabs that remained alive at the end of the experiment had an increase of 0.28 ± 0.23 °Bx (T test, T_142_ = −2.773, *p* = 0.0063). Because the BRIX level was related to mortality, the effect of treatment type on the change in BRIX values was analysed separately for crabs that survived the entire period and those that died during each treatment. There were no significant differences in the magnitude of change of BRIX among the six treatments (comparing initial value and last measured value) for the crabs that perished during the experiment (one way ANOVA, F_63,5_ = 0.883, *p* = 0.498), nor for crabs that survived the 27 week experiment (one way ANOVA, F_69,5_ = 0.962, *p* = 0.447). 

### 3.6. Righting Time

Righting time was typically quite rapid (<5 s); it was influenced by the position and number of limbs lost and these crabs were excluded from the analyses (see methods). The righting time remained relatively constant between mean levels of 1.8 ± 0.2 and 2.6 ± 0.3 s during the first 11 weeks (Figure 5 and Figure 8A). Thereafter, there was a sharp increase in righting times which remained elevated at between 5.5 ± 1.1 and 9.9 ± 2.1 s for the remainder of the experiment. It was noteworthy that during the experiment 22 crabs took an extended time to right (>60 s); of these 16 (73%), died within a week of these high readings. 

### 3.7. Leg Flare

All the crabs exhibited a strong leg flare response in all treatments during the first week of testing (Figure 6). A small number (<5) showed a weak response after two weeks (Figure 8E). However, it was not until 8 to 11 weeks in captivity that a significant increase in the number of crabs with a weak leg flare response (relative to initial levels) occurred. There was a further change at 21 weeks which was reflected by an increase in the overall number of crabs (all treatments combined) showing no leg flare response (Figure 8E).

### 3.8. Leg Retraction

The leg retraction response was significantly affected by the presence of a shelter (Table 3). There was also a significant interaction between shelter and density which was largely driven by single crabs with a shelter exhibiting a greater percentage of strong responses throughout the 27 week period, whereas in all other treatments leg retraction became weaker over time (Figure 7). 

The overall leg retraction response (treatments combined) changed over time, with a steady increase in animals exhibiting a weak response or no response (Figure 8F). The change from initial values was most noticeable after 8 weeks in captivity where an increasing percentage of animals exhibited a weak retraction response. There was a further change at 19 weeks reflected by an increase in the number of crabs exhibiting no response. 

## 4. Discussion

While there were no differences in mortality rates of green crabs, *Carcinus maenas*, as a function of stocking density or presence of shelter, there were some significant differences among treatments for the indicators of vitality. The presence of shelter reduced the number of limbs lost by individual crabs (single and 4 crab treatments). Shelter also affected leg retraction, but this response had an interactive effect with density and only appeared to affect single crabs with a shelter. Because the effects of shelter were not consistent across all vitality indices, this may simply be a chance finding. The current literature also reports discrepancies in the effectiveness of habitat enrichment in aquariums [65,66,67]. The absence of a clear effect of shelter here may be related to the actual type or quality of the shelter [68,69]. In nature, shelter is important for *C. maenas* to escape from predators and avoid bright light [70], and in the intertidal zone they tend to be found beneath seaweed and rocks [71]. The shelters did allow *C. maenas* to avoid bright light, although they were relatively large in relation to the crab and so the crabs may not have been able to push-up the top of the carapace and make contact (thigmotaxis) with the shelter. Such positive thigmotaxis may be a more important cue when seeking a shelter than escape from bright light [72,73]. 

There were no significant effects of stocking density on any of the vitality indices. For other decapod crustaceans inappropriate stocking density can have a negative effect on survival and growth [50,74]. *C. maenas* is a gregarious species often occurring in high densities [71,75] and thus maintaining them in fairly high densities probably does not cause undue stress. The crabs were also fed to excess and so there was no limitation of energy/nutrient intake due to competition for food; an important consideration for stocking density in aquaculture operations or for wild populations [76]. Although shelter and stocking density did not have a consistent effect, there was a clear effect of time (number of weeks crabs were held in the tanks), with an increase in mortality and a decline in vitality, the use of these indices is discussed below.

In decapod crustaceans, limb loss is primarily associated with encounters with predators or conspecifics [77]. In *C. maenas* limb loss occurs in 4% [78] to over 50% [79] of individuals in wild populations. In the current experiment, predators were absent, and so limb losses were likely due to interactions with conspecifics. As shelter did reduce the number of limbs lost in some treatments, one might argue that the crabs were able to avoid interactions with conspecifics (and hence limb loss) by retreating to a shelter. However, in the single crab treatment without a shelter there were no other crabs, and yet this group had the highest limb loss. In addition, in the highest density (8 crabs), where negative interactions were likely to be highest, the presence of shelter did not appear to reduce the number of limbs lost. Therefore, other factors must be responsible. It could be that the crabs were getting their legs caught in the mesh of the cages, causing them to drop the limb [80], and those without shelter would be in contact with the outside of the cage more often. In addition, limb autotomy occasionally occurred during the handling process or when the needle was inserted to take a haemolymph sample; however, there were no differences in handling of the crabs in the different treatments. Thus, it is not clear why the single crabs without a shelter lost more limbs than those in other treatments. 

Because there was no significant difference in the number of limbs lost between crabs that died during the experiment and those that survived, it does suggest that limb loss is not tightly linked with mortality nor an effective indicator of pending mortality. Indeed, several crabs that lost no limbs died during the experiment, while one crab that lost all limbs was still alive at week 27. This is contrary to findings for other species where limb loss does affect survival [reviewed in 77]. However, the individual *C. maenas* used here were not food limited, neither were they in the presence of predators, both of which can influence survival for crustaceans that have lost limbs [77,81]. It is important to note in the context of animal care that autotomy is a natural process whereby an individual sheds a limb. In natural decapod crustacean populations between 2% and 80% of individuals can be missing one or more limbs [77]. One might need to be vigilant if a large number of captive animals are losing multiple limbs, but random limb loss should not be an undue concern for animal care committees. 

The time that *C. maenas* was maintained in the laboratory influenced muscle strength; the reduction recorded here appeared to be a continuous process because the crabs that survived the 27 week period showed a greater decline in claw strength index than those that perished during the experiment. Similar findings are reported for captive Dungeness crabs (*Metacarcinus* (as *Cancer*) *magister*) where a deterioration in individual muscle fibres with delineation of individual sarcomeres occurs after several months in the lab (Dr. Graeme Taylor, pers. comm.). The crabs in this experiment were fed shucked mussels, fish and kelp (soft prey items). When crabs do not use the chelae to open hard shelled prey there tends to be a deterioration of muscle mass, reduction in carapace thickness and crushing ability of the chelae [55,82], and this may be what happened in our experiment. Whether this decline in chelal strength would have any longer-term health implications or could be a useful indicator of time in captivity warrants further investigation.

Overall, there was a transient increase in BRIX levels during the first 3 months, after which they slowly declined back to initial levels. At the start of the experiment the crabs had only been in the lab for 2 weeks, and prior to this (in the wild) their feeding events would have been infrequent and the quality of prey items varied [83,84,85]. In the lab, the crabs were fed a high quality diet and allowed to eat to satiation, which would lead to an increase in haemolymph protein levels [31]. One might expect that because food was not limited that BRIX levels should remain high throughout the experimental period [31]. However, the other vitality indices indicated a deterioration between 8–11 weeks. It is likely that crabs that became “sick” or impaired fed less, and the lowered BRIX level was an indicator of a lower food consumption, rather than the actual cause of mortality [31]. In support of this, the BRIX levels of crabs that died declined slightly, while those animals that were alive at the end of the experiment showed a slight increase in BRIX. The crabs were fed to excess twice per week, so the availability of nutrients was not a limiting factor causing the decline [31]. The decline in BRIX in crabs that died was also unlikely to be the actual cause of death. *C. maenas* can survive for over 3 months without food [86], and exhibit much lower haemolymph protein levels (Rivers and McGaw unpub. obs.) without ill-effect. The use of BRIX and/or haemolymph protein levels is now gaining acceptance in the field of aquaculture as an effective and rapid, relatively non-invasive method to determine the health and vitality of aquacultured crustaceans [32,33,60] and could be an important tool in monitoring crustacean welfare in the laboratory.

A high percentage of individuals exhibiting an increased righting time (>60 s) died before the next measurement (2 weeks), suggesting that righting time is a good indicator of vitality. Righting time also increased over the experimental period, with a sharp inflection after 11 weeks which matches an increased overall mortality of crabs at this time. The increase in righting time appears to be due to an overall loss of muscle strength and coordination because the proportion of crabs exhibiting impairment of leg flare and retraction also increased at this time. 

Overall the changes in the vitality indices became most noticeable after 8–11 weeks in captivity for tagged (handled) crabs. What this time period represents is unclear at present, but because survival of untracked, unhandled crabs was not affected to a similar degree, it suggests that the experimental manipulation and handling, rather than the diet or the actual experimental stocking conditions was a key factor. 

The higher mortality rate of the handled crabs compared with the unhandled (extra) crabs (67% versus 4%) was an incidental, but striking finding of this experiment. The actual emersion of the crabs during vitality tests was unlikely to affect survival because crabs were only emersed once every two weeks for approximately 30 min. *C. maenas* is an effective bimodal breather with aerial oxygen consumption rates ranging between 50% and 120% of those measured in water [87,88], and during short-term aerial exposure (without handling) *C. maenas* does not appear to undergo anaerobic respiration [89]. Aerial exposure does produce an acute stress response (<6 h), indicated by an increase in haemolymph glucose levels, but this is associated with the physical handling of the crabs during transfer, rather than the actual emersion [26]. Aside from recording the vitality indices, the crabs were also handled twice weekly when inspected for mortality and limb loss: this took place underwater and took just seconds, thereby only inducing a short-term, transient stress response [26]. The only other difference between the handled and unhandled crabs was the monthly collection of a 500 µL haemolymph sample. Decapods have an estimated haemolymph volume of between 25 and 35% of their wet mass [90,91], meaning (depending on crab size) approximately 1.5% to 4% of the haemolymph was removed each time. This is a comparatively small amount, and crabs can regenerate this haemolymph volume within days [92]. Indeed, the fact that there was an overall increase in BRIX values for crabs that survived the entire experiment suggests that they are regenerating the haemolymph and its components and it is not becoming diluted. We did not measure bacterial load in the tanks, but tank walls often support an increased load of bacteria [93,94], and the needle puncture site could have been an entry point for opportunistic bacterial infections [95]. 

While the exact reason for the increased mortality of handled crabs remains conjecture, it is unlikely due to a single factor, but rather a multitude of factors accumulating over time [96,97]. In crustaceans, repeated short-term stress causes the release of hormones that leads to a chronic stress response [96]; thus even though the experimental measurements were infrequent, they could have longer-term cumulative effects. This is known as an “allostatic load,” whereby repeated acute stress responses may have chronic downstream effects [98]. These continued periods of stress likely exhaust energy stores: white shrimp, *Litopenaeus* spp., subjected to chronic stress exhibit lower levels of haemolymph protein, total lipids and triglycerides and reduced total haemocyte counts [20,99] leaving them vulnerable to common bacterial infections [95]. The high mortality rate for handled *C. maenas* is important to consider because most captive crustaceans are not simply being held for scientific display or exhibit, but rather are used in specific experiments. Most of these experiments will involve handling, transfer between areas, or some type of treatment. If these handling procedures can cause a significant and noticeable increase in mortality rates, this is something that must be considered when performing experiments on captive-held crustaceans: handling procedures should be kept to a minimum when planning long-term, repeated measures experiments. Given this relatively high loss of crabs, when applying the three R’s (reduction, refinement, replacement), one may need to obtain and hold a higher initial number of animals than would be appear necessary for the actual experimental protocols [66].

## 5. Conclusions

A noticeable decline in vitality and an increase in mortality in captive *C. maenas,* occurred between 8 and 11 weeks, suggesting holding times longer than 3 months may become problematic. Whilst we acknowledge that each species is likely to have a different optimum laboratory holding environment due to their differences in physiology and behaviour, the use of *C. maenas* provides a baseline ‘minimum requirement’ for the stocking of less hardy species. It will be important to gather similar data from other species from different environments and degrees of physiological and stress tolerance to determine any potential problems in housing decapod crustaceans for long term periods in the laboratory. As Crustacea represent a new taxon for animal care regulation, this type of baseline data is essential to help inform and educate in the preparation of animal care protocols and their subsequent review by the committees.

## Figures and Tables

**Figure 1 animals-12-02970-f001:**
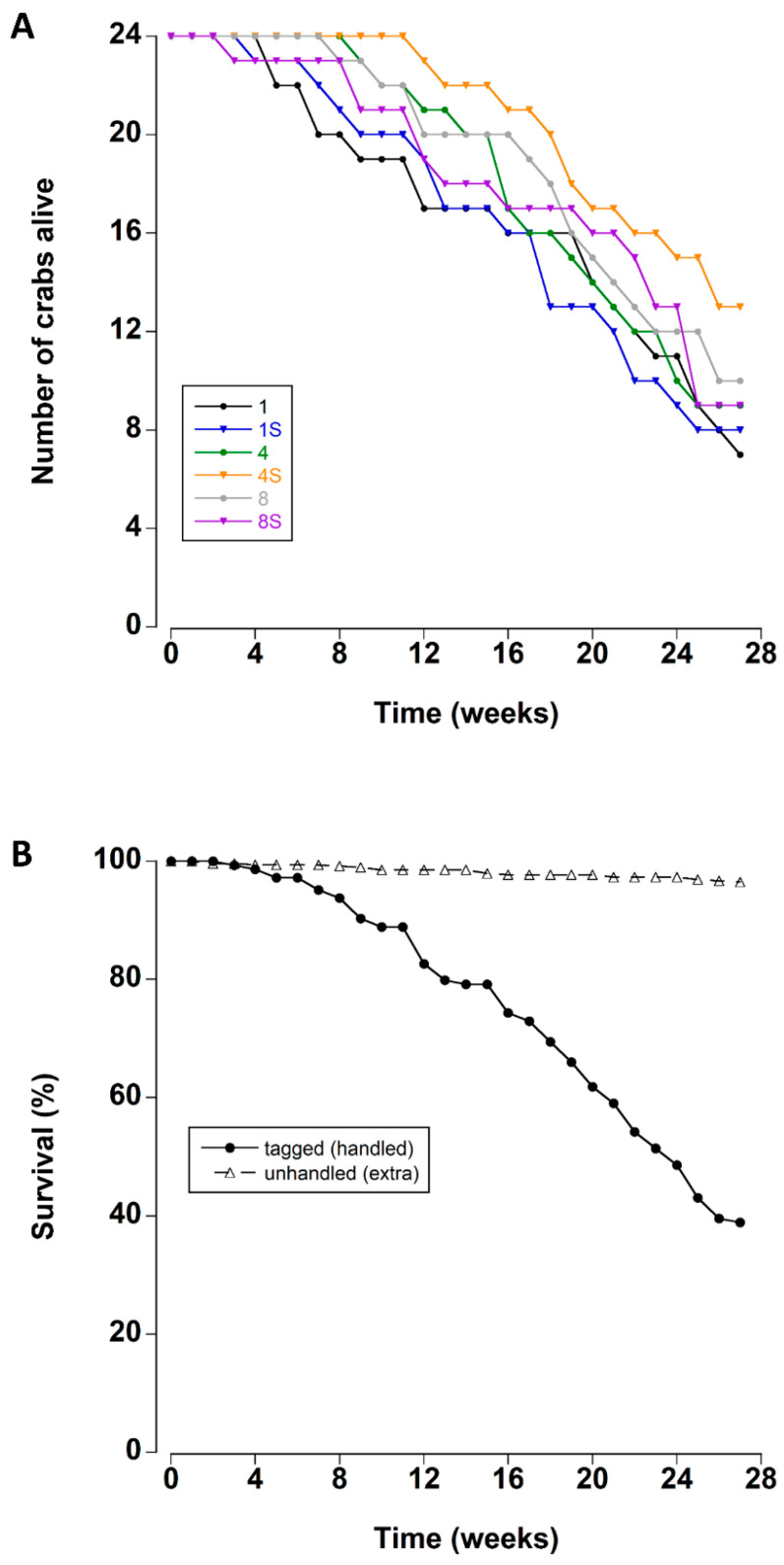
(**A**) Survival rates of adult male *C. maenas* maintained in the laboratory in densities of 1, 4 or 8 crabs, with (S) or without shelter (*n* = 24 per treatment at start of experiment) (**B**) Survival rates (expressed as a percentage) of experimental crabs (tagged/handled) and extra (unhandled, additional animals to maintain density) crabs during a six month period in the laboratory.

**Figure 2 animals-12-02970-f002:**
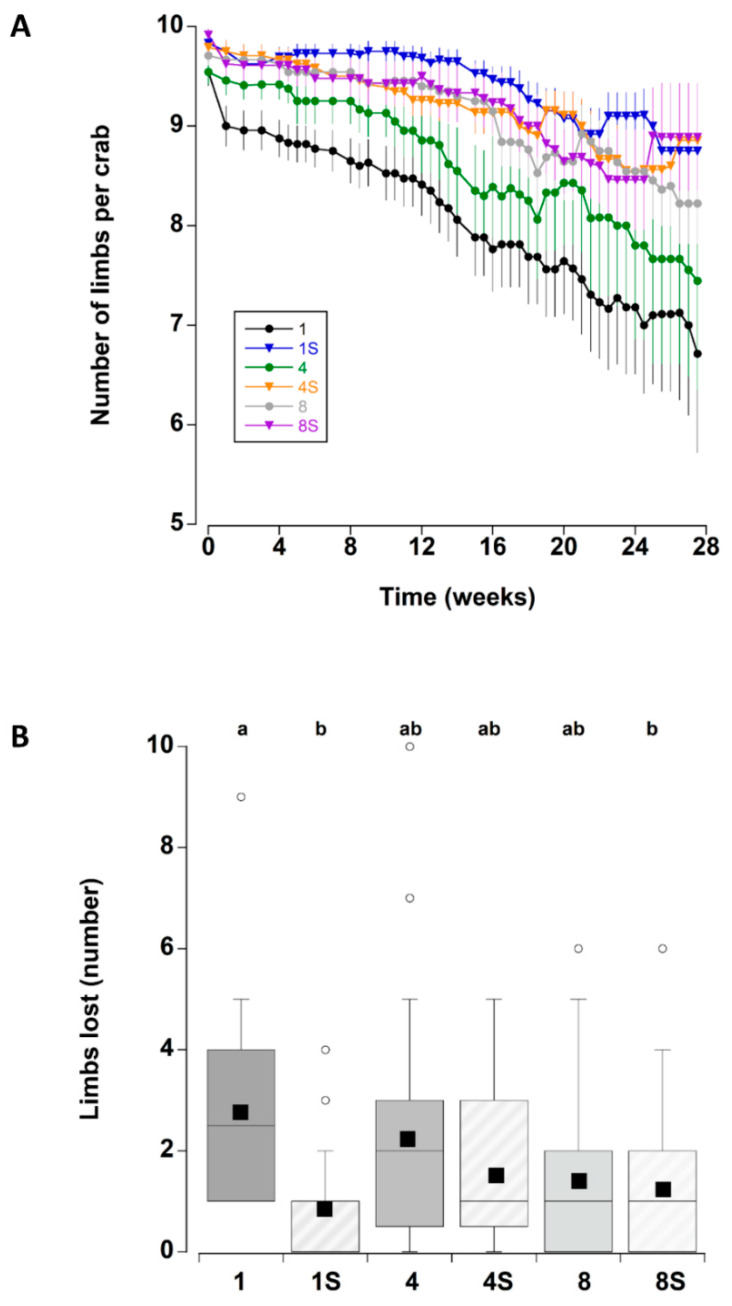
(**A**) Number of limbs per adult male *C. maenas* (mean ± SE) maintained in the laboratory in densities of 1, 4 or 8 crabs, with (S) or without shelter (*n* = 24 crabs per treatment). As the crabs lost limbs the number remaining per crab decreased; in some cases there is an apparent rise associated with the death of a crab and this animal being removed from subsequent calculations of mean numbers (**B**) Total number of limbs lost by individual *C. maenas* in the six different treatments during the six month experimental period. If a crab died during the experiment the number of limbs lost prior to death was recorded. The box plots represent the 75% confidence intervals, whiskers indicate the 95% confidence limits of the data, the horizontal line is the median number and the square is the mean number of limbs lost. Different letters above each box represent significant differences (*p* < 0.05).

**Figure 3 animals-12-02970-f003:**
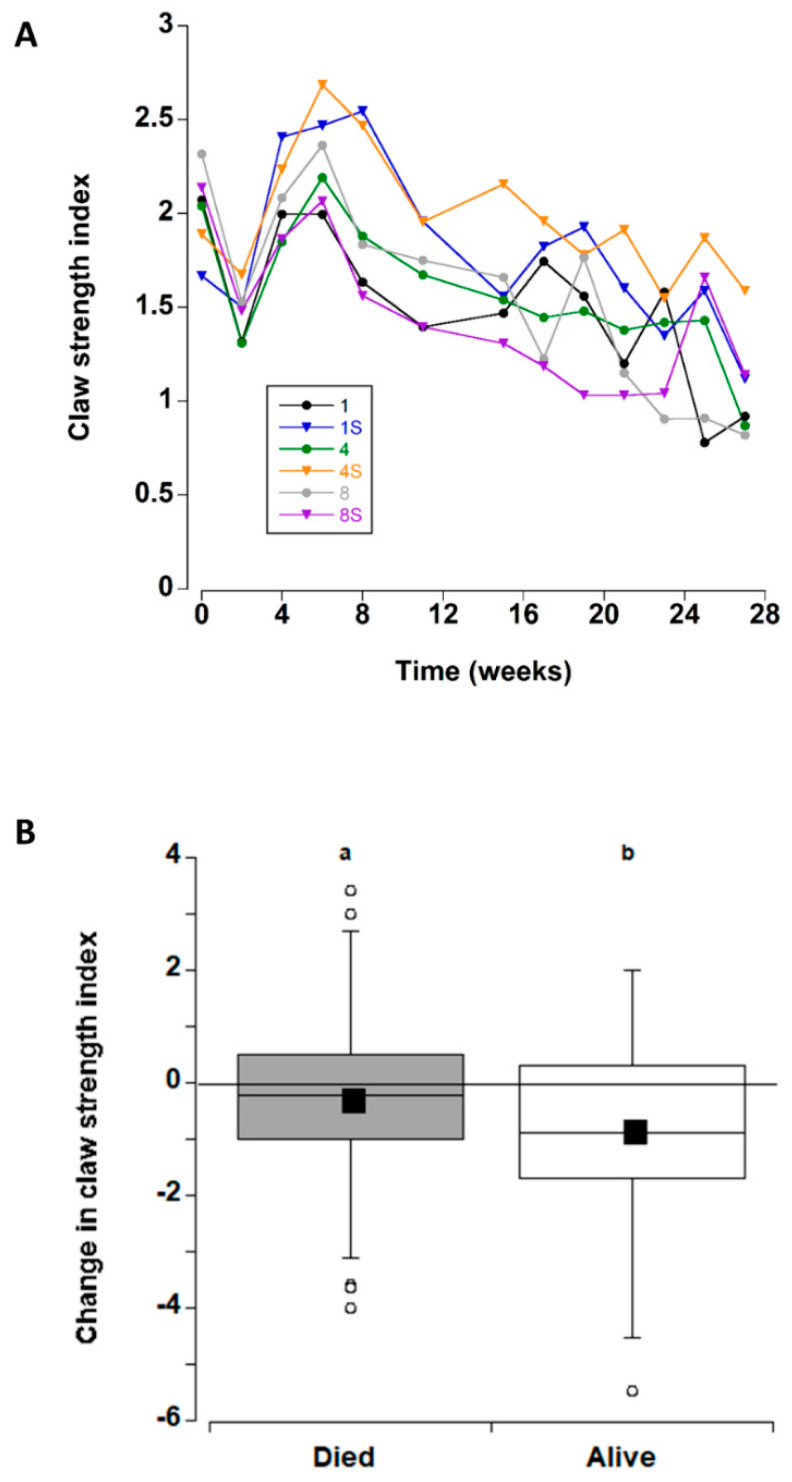
(**A**) Claw strength index values (strength/propodus height) for *C. maenas* (mean ± SE) kept in densities of 1, 4 or 8 crabs, with (S) or without shelter (*n* = 24 per treatment) for six months in the laboratory (**B**) Change in claw strength index (between first and last measurement) for individual crabs. If a crab died or lost/broke a claw during the experiment, the last measured value was used. In this case, the changes are shown between crabs that died/ceased to be recorded during the experiment and those that survived during the experiment (data for the 6 different treatments was combined). The box plots represent the 75% confidence intervals, whiskers indicate the 95% confidence limits of the data, the horizontal line is the median number and the square is the mean number of limbs lost. Different letters above each box represent significant differences (*p* < 0.05).

**Figure 4 animals-12-02970-f004:**
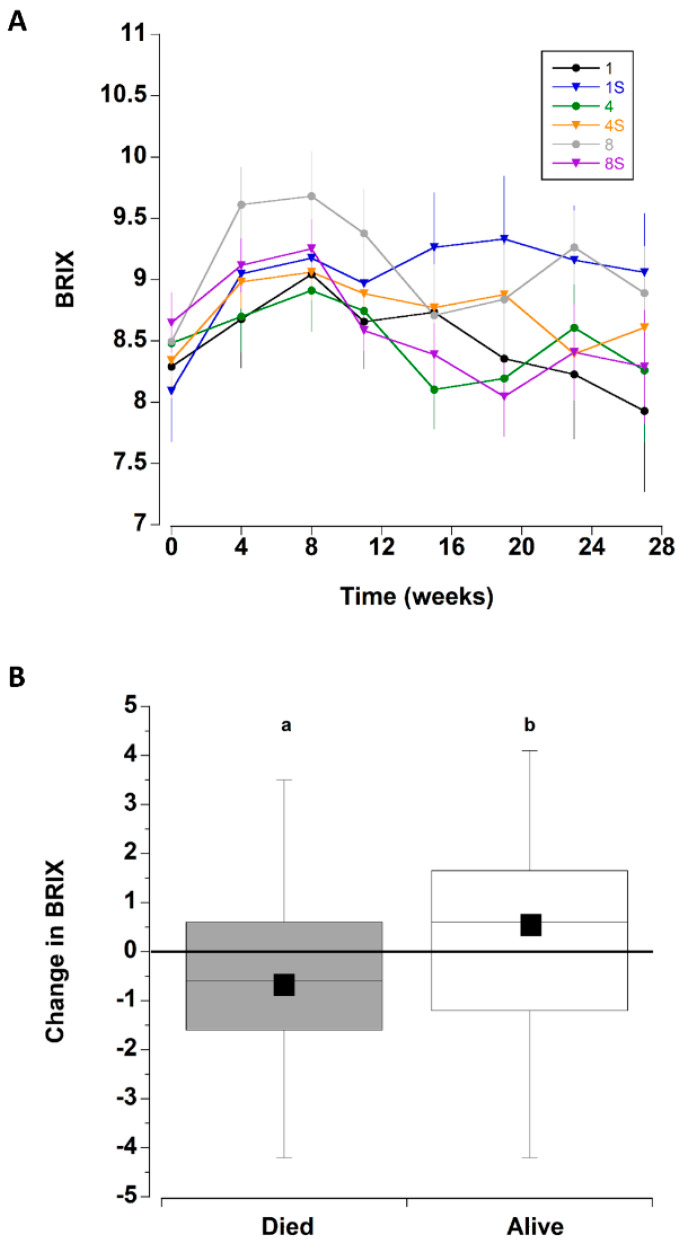
(**A**) BRIX values for *C. maenas* (mean ± SE) kept in densities of 1, 4 or 8 crabs, with (S) or without shelter. (*n* = 24 per treatment) for six months in the laboratory. (**B**) Change in BRIX values (between first and last measurement) for crabs that died during the experiment (where the last measured value was used) and those that survived the six month experimental period. Data for the 6 different treatments was combined. The box plots represent the 75% confidence intervals, whiskers indicate the 95% confidence limits of the data, the horizontal line is the median number and the square is the mean number of limbs lost. Different letters above each box represent significant differences (*p* < 0.05).

**Figure 5 animals-12-02970-f005:**
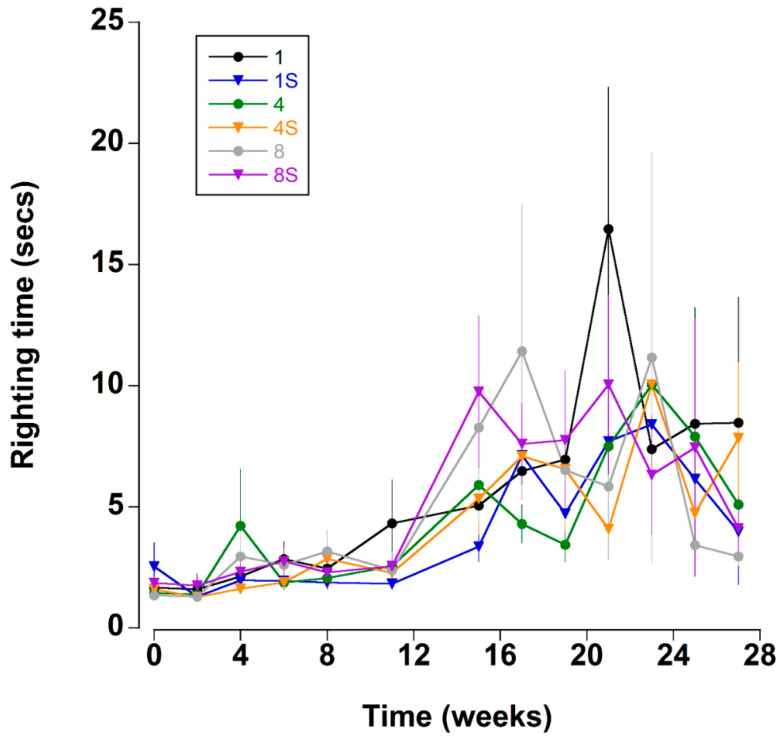
Righting time (s) (time taken to move from a ventrum up to dorsum up position) of *C. maenas* (mean ± SE) kept in densities of 1, 4 or 8 crabs, with (S) or without shelter (*n* = 24 per treatment).

**Figure 6 animals-12-02970-f006:**
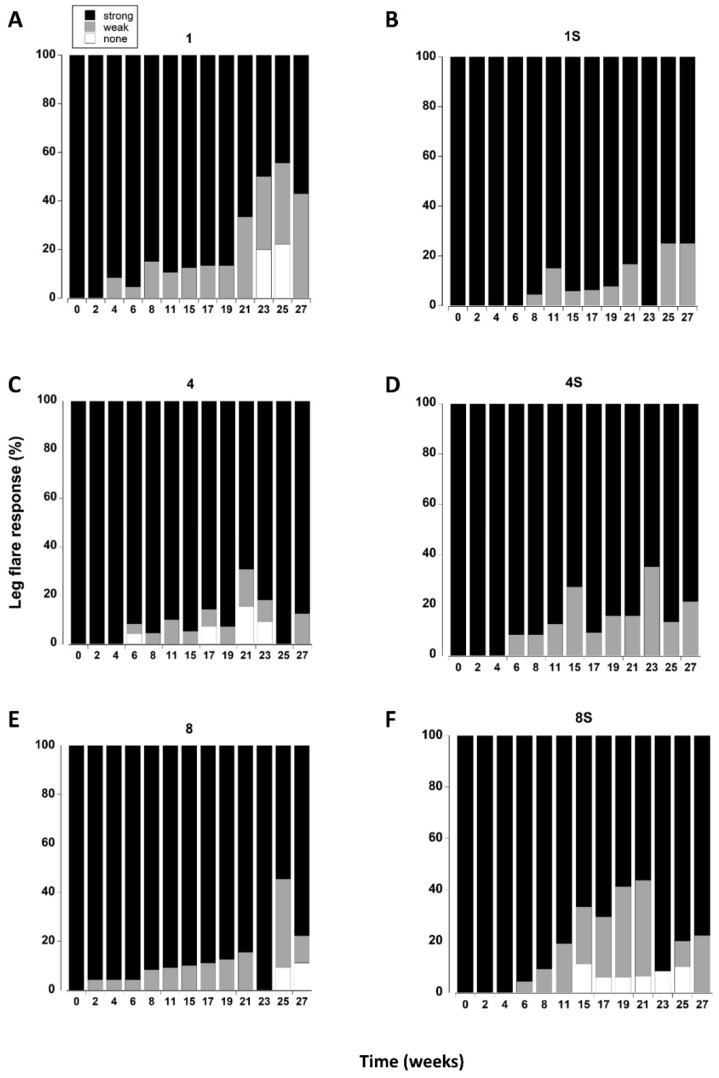
(**A**–**F**). Leg flare responses of *C. maenas* (mean ± SE) kept in densities of 1, 4 or 8 crabs, with (S) or without shelter. The flare response was classified as strong, weak or absent (Table 1) and expressed as a percentage of animals alive at each time period. If a crab died or lost both of its fourth pair of walking legs, or lost > 3 legs on one side it was not included in the measurements.

**Figure 7 animals-12-02970-f007:**
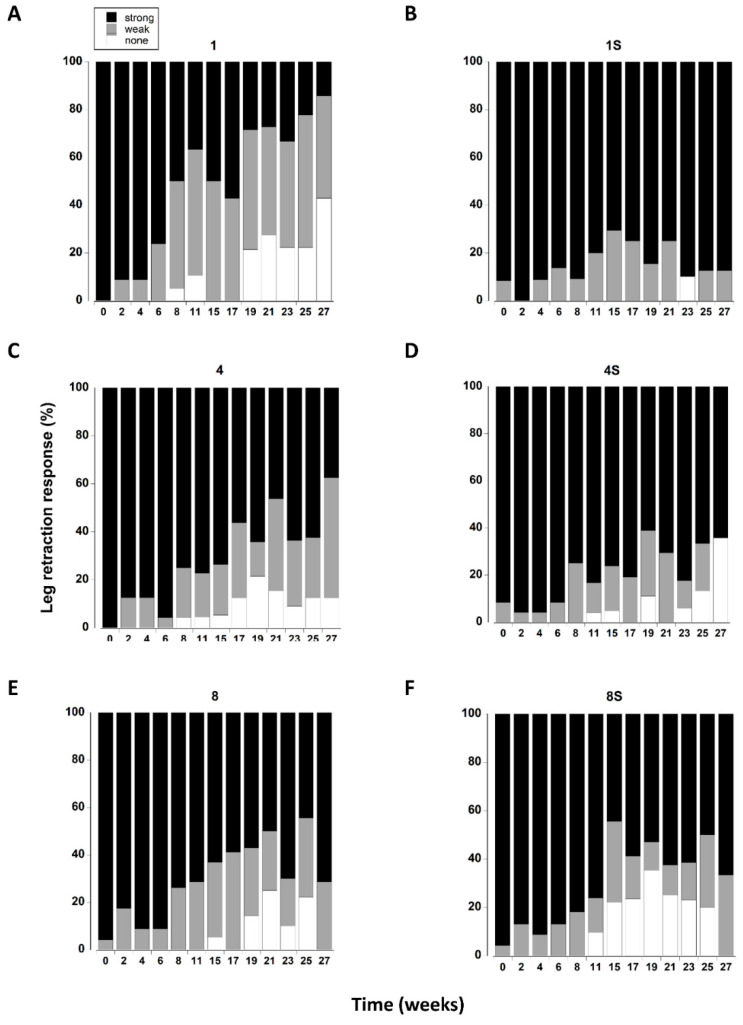
(**A**–**F**) Leg retraction responses of *C. maenas* (mean ± SE) kept in densities of 1, 4 or 8 crabs, with (S) or without shelter. The retraction response was classified as strong, weak or absent (Table 1) and expressed as a percentage of animals reaming alive at each time period. If a crab died or lost both of its first pair of walking legs, or lost > 3 legs on one side it was not included in the measurements.

**Figure 8 animals-12-02970-f008:**
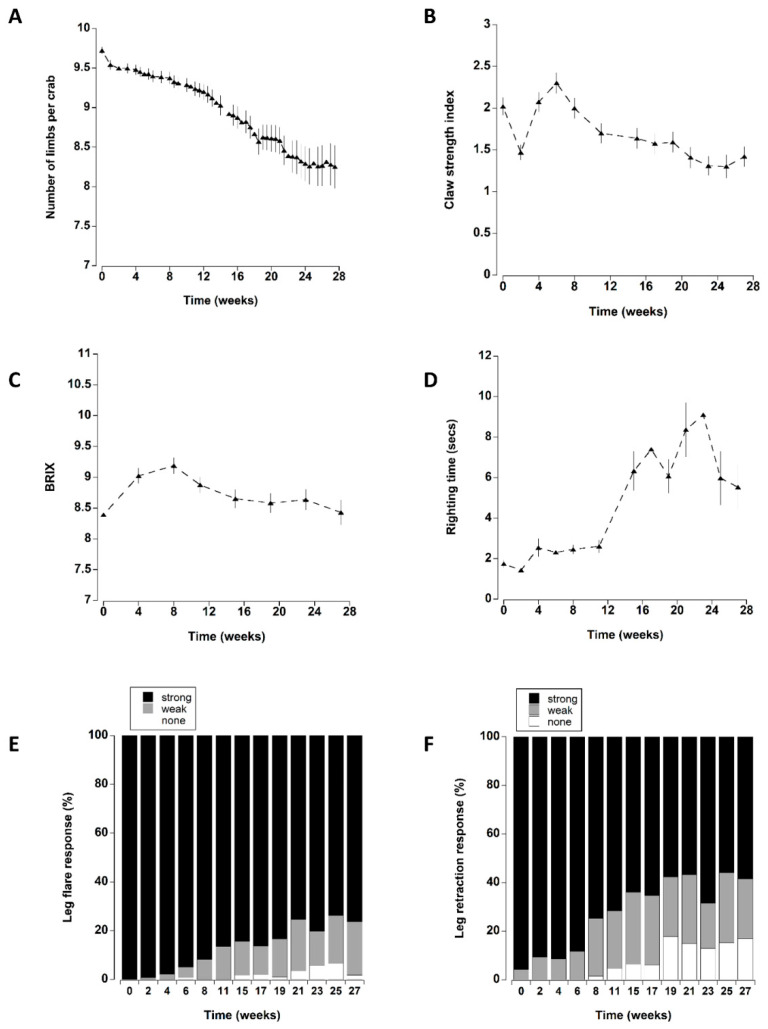
Data for all six treatments (density of 1, 4, 8 crabs, with or without shelter) combined to show changes over time. (**A**) Number of limbs remaining per crab. (**B**) Claw strength index (**C**) BRIX values and (**D**) righting time (seconds): data for these four graphs represent the mean ± SE. (**E**) Leg flare and (**F**) Leg retraction expressed as percentage of animals (remaining alive) exhibiting a strong, weak or no response.

**Table 1 animals-12-02970-t001:** Behavioural reflex reaction indicators used to quantify *C. maenas* vitality over time. Measurements were taken every two weeks and recorded as either strong, weak or no response. Adapted from [35].

Reflex	Test	Strong Response	Weak Response	No Response	No Longer Used for Testing If:
**Leg flare**	Crabs gently lifted out of water by the carapace, dorsum up	All walking legs spread wide and high. Back legs can be extended higher than horizontal	Partial response, some legs remain below horizontal	No attempt to flare legs, all appendages remain below horizontal	Loss of both right and left 4th walking legs, 3 or more limbs on one side
**Leg retraction**	Crabs gently lifted out of water, dorsum up. Attempt to manually retract the first walking leg anteriorly	Crabs resist to the motion–leg stays in its current position	Crabs show less resistance, but leg always returns to original position after manipulation	No resistance from crab, and leg droops down.	Loss of both left and right first walking legs, or loss of more than three limbs on one side

**Table 2 animals-12-02970-t002:** Mean (±SE) and median survival times for *C. maenas* maintained for 6 months at densities of 1, 4, and 8 crabs, with or without the addition of shelter (*n* = 24 per treatment).

Treatment	Mean ± SE (weeks)	Median (weeks)
1 crab	19.6 ± 1.7	22
1 crab + shelter	19.4 ± 1.6	22
4 crabs	21.2 ± 1.3	24
4 crabs + shelter	23.6 ± 1.0	-
8 crabs	21.8 ± 1.3	26
8 crabs + shelter	21.0 ± 1.5	25

**Table 3 animals-12-02970-t003:** Statistical analysis of the vitality indices using full factorial model with main effects for time, shelter, density and their various interactions. Using a non-parametric analysis for longitudinal data (nparLD), ANOVA-type F (F statistic) and *p* values and the degrees of freedom (DF) are given for each factor and interactions. Significant results (*p* < 0.05) are in bold.

Vitality Index	F Statistic	DF	*p* Value
** *Limb Loss* **			
Density	0.8	2	0.437
**Shelter**	**10.4**	**1**	**0.001**
**Time**	**30.4**	**47**	**<0.000**
**Density × Shelter**	**3.0**	**2**	**0.047**
Density × Time	0.8	94	0.534
Shelter × Time	0.8	47	0.487
Density × Shelter × Time	0.8	94	0.564
** *Claw Strength* **			
Density	1.0	2	0.374
Shelter	1.3	1	0.289
**Time**	**11.3**	**10**	**<0.000**
Density × Shelter	1.3	2	0.269
Density × Time	1.3	20	0.237
Shelter × Time	0.7	10	0.647
Density × Shelter × Time	0.9	20	0.538
** *BRIX* **			
Density	0.3	2	0.756
Shelter	0.0	1	0.908
**Time**	**7.5**	**6**	**<0.000**
Density × Shelter	1.4	2	0.248
Density × Time	1.1	12	0.376
Shelter × Time	0.6	6	0.601
Density × Shelter × Time	1.6	12	0.164
** *Righting Time* **			
Density	0.7	2	0.51
Shelter	1.5	1	0.222
**Time**	**70.6**	**12**	**<0.000**
Density × Shelter	0.9	2	0.414
Density × Time	0.7	24	0.808
Shelter × Time	1.1	12	0.395
Density × Shelter × Time	0.5	24	0.902
** *Leg Flare* **			
Density	0.4	2	0.666
Shelter	0.0	1	0.94
**Time**	**8.0**	**12**	**<0.000**
Density × Shelter	2.5	2	0.088
Density × Time	1.7	24	0.083
Shelter × Time	1.1	12	0.355
Density × Shelter × Time	1.5	24	0.121
** *Leg Retraction* **			
Density	0.6	2	0.525
**Shelter**	**8.9**	**1**	**0.003**
**Time**	**15.4**	**12**	**<0.000**
**Density × Shelter**	**5.2**	**2**	**0.006**
Density × Time	0.9	24	0.579
Shelter × Time	2.0	12	0.053
Density × Shelter × Time	1.3	24	0.192

## Data Availability

Data presented in this study are available upon request from ijmcgaw@mun.ca.

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
