# Peer review of "Effect of Animal Stocking Density and Habitat Enrichment on Survival and Vitality of Wild Green Shore Crabs, *Carcinus maenas,* Maintained in the Laboratory"

_animals, 2022, doi:10.3390/ani12212970_

Round 1
Reviewer 1 Report
Review
Paper title: Effect of animal stocking density and habitat enrichment on survival and vitality of wild green shore crabs, Carcinus maenas, maintained in the laboratory
The authors conducted a laboratory study to reveal the effects of rearing conditions such as stocking density, habitat enrichment, and handling on the survival and vitality of wild green shore crabs. The authors found that stocking density also had no effects on the vitality indices and survival rates. The presence of shelter did affect survival and some vitality indices but changed significantly the number of limbs lost and the leg retraction response. The authors registered an increase in mortality rates and argued this result was associated with multiple handling. This study highlights the importance of maintaining suitable rearing conditions in scientific experiments.
All these reasons explain the relevance of the paper by Charlotte H. Wilson and co-authors submitted to "Animals".
General scores.
The data presented by the authors are original and significant. The study is correctly designed and the authors used appropriate sampling methods. In general, statistical analyses are performed with good technical standards. The authors conducted careful work that may attract the attention of a wide range of specialists focused on crustacean aquaculture and welfare.
Recommendations.
The authors should add a "Simple summary" section according to Rules for Authors.
The authors should check and correct the numbering of figures. Currently, Fig. 1 is followed by Fig. 8.
Fig. 8E, F. The authors should increase the font size and resolution for the legends.
Fig. 3A, 4A. The authors should specify the legend: what do mean "B, D, F,…L".
Citations and references should be formatted according to Rules for Authors.
Specific remarks.
L 47. Consider replacing “makes them” with “make them”
L 75. Consider replacing “The measurement of acute stress responses are” with “The measurement of acute stress responses is”
L 111. Consider replacing “has made it” with “have made it”
L 135. Consider replacing “crabs fed” with “crabs were fed”
L 145. Consider replacing “width measured and any appendage loss” with “width was measured and any appendage loss was”
L 147. Consider replacing “a carapace width” with “a carapace width of”
L 168. Consider replacing “mortalities” with “mortalities were”
L 177. Consider replacing “from their original tank” with “from its original tank”
L 238. Consider replacing “reading obtained” with “reading was obtained”
L 272. Consider replacing “a Students t test” with “Student's t-test”
L 454. Consider replacing “experiment t” with “experiment”
L 482. Consider replacing “shelter . .” with “shelter.”
L 510. Consider replacing “discrepancies on” with “discrepancies in”
L 531. Consider replacing “are discussed” with “is discussed”
L 541. Consider replacing “the numbers” with “the number”
L 665. Consider replacing “As crustaceans represent a new taxa” with “As Crustacea represents a new taxon”
Author Response
The data presented by the authors are original and significant. The study is correctly designed and the authors used appropriate sampling methods. In general, statistical analyses are performed with good technical standards. The authors conducted careful work that may attract the attention of a wide range of specialists focused on crustacean aquaculture and welfare.
Recommendations.
The authors should add a "Simple summary" section according to Rules for Authors.
added
The authors should check and correct the numbering of figures. Currently, Fig. 1 is followed by Fig. 8.
We have reworded in some areas. Figure 8 shows all the data as a summary (with shelter and density treatments removed). In each section where we do mention Fig 8 we have added overall. In the Vitality Indices section we do list Figures 2-7 first and then figure 8
Fig. 8E, F. The authors should increase the font size and resolution for the legends.
This has been done
Fig. 3A, 4A. The authors should specify the legend: what do mean "B, D, F,…L".
Sorry this was a mistake and has been corrected
Citations and references should be formatted according to Rules for Authors.
Done
Line numbers on our original copy nor the one provided by MDPI actually line up so hopefully we can find these
Specific remarks.
L 47. Consider replacing “makes them” with “make them”
changed
L 75. Consider replacing “The measurement of acute stress responses are” with “The measurement of acute stress responses is”
Changed
L 111. Consider replacing “has made it” with “have made it”
Changed
L 135. Consider replacing “crabs fed” with “crabs were fed”
Added
L 145. Consider replacing “width measured and any appendage loss” with “width was measured and any appendage loss was”
Was added
L 147. Consider replacing “a carapace width” with “a carapace width of”
Of added
L 168. Consider replacing “mortalities” with “mortalities were”
Changed to any mortalities were
L 177. Consider replacing “from their original tank” with “from its original tank”
changed
L 238. Consider replacing “reading obtained” with “reading was obtained”
Was added
L 272. Consider replacing “a Students t test” with “Student's t-test”
Changed
L 454. Consider replacing “experiment t” with “experiment”
This was in legends and it has been removed
L 482. Consider replacing “shelter . .” with “shelter.”
Because reference to specific lines do not match up we were unable to find exactly where this was
L 510. Consider replacing “discrepancies on” with “discrepancies in”
changed
L 531. Consider replacing “are discussed” with “is discussed”
changed
L 541. Consider replacing “the numbers” with “the number”
Changed
L 665. Consider replacing “As crustaceans represent a new taxa” with “As Crustacea represents a new taxon”
Changed
Reviewer 2 Report
Effect of animal stocking density and habitat enrichment on survival and vitality of wild green shore crabs, Carcinus maenas, maintained in the laboratory by Wilson et al . animals-1968940.
The study deals with the effects of stocking density and shelter on survival and vitality indices of the green shore crab, Carcinus maenas during a 6 month period in unnatural conditions associated with the laboratory environment. This interesting work helps to known about how captive conditions affect the survival and general condition of wild decapod crustaceans. It also provides an important baseline information on the responses to long-term laboratory captivity, from which optimal storage conditions for other species could be refined. I personally appreciate the paper at this time when decapod crustaceans and marine animals are being introduced into captivity and extensively used in research related to environmental changes. The introduction of the manuscript was appropriate to the presented hypothesis. The methodology is carefully designed and presented, but the results can still be improved in their presentation. The Discussion and conclusions are consistent with the obtained findings. The text is well written and clear for understanding. Authors should consult the MDPI Guide for Authors for improving author citations in the text as well as in the reference lists. In my opinion the manuscript is acceptable for publication in the Animals after minor revision. I have only minor suggestions/comments as listed below:
L133 What meant by « S = 31-32 » ? See also L164. It’s unclear. Please improve.
L290-292 In the legend to Figure 1B, authors should clearly explain the terms “tracked” and “extra” used in Figure (1B) for quick understanding without referring to the main text.
L308-363 I suggest that the authors split Table 3 into 2 or 3 smaller tables and improve their presentation. For example Table 3a with vitality index of Lomb Loss, Claw Strength and BRIX and Table 3b with Righting Time, Leg Flare and Leg Retraction.
L364 What meant by « (s)» of the righting time? See Fig. 8D and caption of Fig. 8 L367.
Please check and correct figure numbering. See Fig. 8 in L365 before Fig. 2 in L384 ?
L383 in Fig. 2B I suggest specifying the unit for the y-axis label
L420 What do the labels B to L shown in fig. 3 claw strength index values? Please explain it in the caption for clarity. L449 See also Figure 4A BRIX values.
Please harmonize the use of labels for the 6 treatments. The use of labels B to L (Fig. 3A, 4A) and 1 to 8S (Fig1A, 2, 5) makes the text messy which makes it difficult to read fluently and understand the paper.
L469 "Righting time (s) and with (S) or without shelter", that's too many "s". Please improve this.
L481 « Figure 6. A-Fc. » = « Figure 6. A-F. »
L493 « Figure 7. A-F) » = « Figure 7. A-F »
Authors should consult the MDPI Guide for Authors for improving author citations in the text as well as in the reference lists (L678-985).
Author Response
Authors should consult the MDPI Guide for Authors for improving author citations in the text as well as in the reference lists. In my opinion the manuscript is acceptable for publication in the Animals after minor revision. I have only minor suggestions/comments as listed below:
L133 What meant by « S = 31-32 » ? See also L164. It’s unclear. Please improve.
In both cases we have changed T to temperature and S to salinity
L290-292 In the legend to Figure 1B, authors should clearly explain the terms “tracked” and “extra” used in Figure (1B) for quick understanding without referring to the main text.
Thank you for spotting this, in the figure legend we changed to tagged and unhandled crabs to match up with the description in the written legend
L308-363 I suggest that the authors split Table 3 into 2 or 3 smaller tables and improve their presentation. For example Table 3a with vitality index of Lomb Loss, Claw Strength and BRIX and Table 3b with Righting Time, Leg Flare and Leg Retraction.
We have kept as one but actually put it into a proper word table
L364 What meant by « (s)» of the righting time? See Fig. 8D and caption of Fig. 8 L367.
S for seconds, we have changed to secs for clarity
Please check and correct figure numbering. See Fig. 8 in L365 before Fig. 2 in L384 ?
In the vitality section we list figures 2-7 and describe the overall changes as figure 8. In each section where we list figure 8 we have added the word ‘overall’ to show we are discussing changes in all crabs collectively
L383 in Fig. 2B I suggest specifying the unit for the y-axis label
We have added (number)
L420 What do the labels B to L shown in fig. 3 claw strength index values? Please explain it in the caption for clarity. L449 See also Figure 4A BRIX values.
This was a mistake –we have added the proper description
Please harmonize the use of labels for the 6 treatments. The use of labels B to L (Fig. 3A, 4A) and 1 to 8S (Fig1A, 2, 5) makes the text messy which makes it difficult to read fluently and understand the paper.
Changed – this was a mistake on our part
L469 "Righting time (s) and with (S) or without shelter", that's too many "s". Please improve this.
We have changed the righting time S which stood for seconds to secs
L481 « Figure 6. A-Fc. » = « Figure 6. A-F. »
corrected
L493 « Figure 7. A-F) » = « Figure 7. A-F »
corrected
Authors should consult the MDPI Guide for Authors for improving author citations in the text as well as in the reference lists (L678-985).
Changed as requested
Additional Items
We have added a statement on animal care
We have updated author contributions in response to the list provided
We have added a short summary statement